

# Growth, stoichiometry and cell size; temperature and nutrient responses in haptophytes

Lars Fredrik Skau[1], Tom Andersen[1], Jan-Erik Thrane[2] and Dag Olav Hessen[1]

[1] Department of Bioscience, University of Oslo, Oslo, Norway
[2] Norwegian Institute for Water Research, Oslo, Norway

## ABSTRACT

Temperature and nutrients are key factors affecting the growth, cell size, and physiology of marine phytoplankton. In the ocean, temperature and nutrient availability often co-vary because temperature drives vertical stratification, which further controls nutrient upwelling. This makes it difficult to disentangle the effects of temperature and nutrients on phytoplankton purely from observational studies. In this study, we carried out a factorial experiment crossing two temperatures (13° and 19 °C) with two growth regimes (P-limited, semi-continuous batch cultures ["−P"] and nutrient replete batch cultures in turbidostat mode ["+P"]) for three species of common marine haptophytes (*Emiliania huxleyi, Chrysochromulina rotalis* and *Prymnesium polylepis*) to address the effects of temperature and nutrient limitation on elemental content and stoichiometry (C:N:P), total RNA, cell size, and growth rate. We found that the main gradient in elemental content and RNA largely was related to nutrient regime and the resulting differences in growth rate and degree of P-limitation, and observed reduced cell volume-specific content of P and RNA (but also N and C in most cases) and higher N:P and C:P in the slow growing −P cultures compared to the fast growing +P cultures. P-limited cells also tended to be larger than nutrient replete cells. Contrary to other recent studies, we found lower N:P and C:P ratios at high temperature. Overall, elemental content and RNA increased with temperature, especially in the nutrient replete cultures. Notably, however, temperature had a weaker–and in some cases a negative–effect on elemental content and RNA under P-limitation. This interaction indicates that the effect of temperature on cellular composition may differ between nutrient replete and nutrient limited conditions, where cellular uptake and storage of excess nutrients may overshadow changes in resource allocation among the non-storage fractions of biomass (e.g. P-rich ribosomes and N-rich proteins). Cell size decreased at high temperature, which is in accordance with general observations.

## INTRODUCTION

Marine phytoplankton constitute nearly 50% of the global primary productivity (*Field et al., 1998*) and are an essential component in the biogeochemical cycling of carbon (C), nitrogen (N), and phosphorus (P) (*Arrigo, 2005*). Increased temperatures in the ocean

Corresponding author
Jan-Erik Thrane,
jan-erik.thrane@niva.no

due to climate change will influence phytoplankton both directly through temperature dependent physiological processes, but also indirectly through increased water column stability. The latter will likely limit nutrient supply and reduce productivity in stratified tropical waters, but may increase productivity at higher latitudes, where phytoplankton growth is constrained by light and deep mixing (*Behrenfeld et al., 2006*; *Boyce, Lewis & Worm, 2010*). Large expanses of the ocean phytoplankton growth is nutrient-limited (*Moore et al., 2013*), hence the interaction between temperature and nutrient status is of considerable ecological and biogeochemical importance. Knowledge of how phytoplankton growth, stoichiometry, and size structure respond to combined changes in these factors is important to predict how productivity, food web dynamics, and the biogeochemical cycling of C, N and P will be affected in a warmer and more stratified ocean (*Arrigo et al., 1999*; *Arrigo, 2005*; *Litchman et al., 2015*; *Sommer et al., 2017*).

Direct effects of temperature on phytoplankton are related to kinetic responses. Temperature increases enzyme and ribosome activity and thereby processes like nutrient uptake and translation. Ultimately, this leads to higher rates of overall processes like protein synthesis, light saturated photosynthesis, and cell division when temperature is elevated towards the optimum and other factors are non-limiting (*Raven & Geider, 1988*; *Davidson, 1991*). Temperature adjustment of growth in autotrophs has also been related to elemental content and stoichiometry since allocation to different macromolecular pools, and thereby the absolute and relative requirements for elements like N and P, may change with temperature (*Shuter, 1979*; *Rhee & Gotham, 1981*; *Sterner & Elser, 2002*). In some phytoplankton species (*Rhee & Gotham, 1981*), among species of higher plants (*Reich & Oleksyn, 2004*), and in ectotherms in general (*Woods et al., 2003*), cellular content of N and P tend to increase in organisms grown at low temperature. N and P do not necessarily change in proportion along temperature gradients (*Reich & Oleksyn, 2004*; *Yvon-Durocher et al., 2015*), and with increasing marine temperatures one prediction is that phytoplankton N:P ratios will increase globally (*Toseland et al., 2013*). This prediction is, in part, based on a "translation compensation hypothesis", which states that because specific ribosomal reaction rate increases with temperature, lower ribosomal density will be needed to maintain the same level of protein synthesis if temperature increases (*Toseland et al., 2013*; *Daines, Clark & Lenton, 2014*). Since ribosomes are rich in P and have a low N:P ratio relative to the cell as a whole, lower ribosome content will increase cellular N:P (*Sterner & Elser, 2002*). Photosynthetic activity—and thereby the allocation to N rich photosynthetic machinery— is also expected to increase with temperature, at least under nutrient replete conditions (*Shuter, 1979*; *Sterner & Elser, 2002*). This may further tend to elevate N:P (and C:P) if temperature increases.

Several observational studies have revealed a positive correlation between the N:P ratio of marine seston and global temperature (*Martiny et al., 2013*; *Toseland et al., 2013*; *Yvon-Durocher et al., 2015*). Yet these large-scale variations are likely mostly related to changes in community composition (*Martiny et al., 2013*; *Yvon-Durocher et al., 2015*), with contributions from spatial differences in dissolved nutrient concentrations (*Galbraith & Martiny, 2015*), geographical differences in the relative contribution of non-algal particles to seseton, and perhaps reduced use of phospholipids in cell membranes in oligotrophic

regions (*Van Mooy et al., 2009*). Not much is known about the contribution from plasticity or temperature acclimation within species.

Also, responses in cell size have been proposed as a common response to elevated temperatures across phyla and taxa (*Atkinson, Ciotti & Montagnes, 2003*; *Daufresne, Lengfellner & Sommer, 2009*; *Gardner et al., 2011*; *Sheridan & Bickford, 2011*). But since small cell size is beneficial under nutrient scarcity (due to higher surface-to-volume ratio), and warming will cause both higher water temperature and generally more oligotrophic conditions through increased stratification, it is difficult to disentangle the drivers based on *in situ* studies, at least for intraspecific changes in cell size. Experimental studies on phytoplankton indicate contributions both from ambient nutrient levels and temperature per se to cell size (*Peter & Sommer, 2013*), although resource availability (productivity) is the dominating driver for the size structure of the community (*Marañón et al., 2012*).

To elucidate stoichiometric and cell size responses to temperature and nutrient limitation, and the potential differences in temperature response between nutrient replete and nutrient limited conditions, we conducted a factorial experiment with three common marine haptophytes crossing two temperatures with two nutrient regimes. We measured the response of growth rate, elemental content (C, N, and P), total RNA content, and cell size, and hypothesized that (1) high temperature will decrease elemental and RNA content, (2) high temperature will increase N:P and C:P ratios, (3) high temperature will reduce cell size, (4) growth rate and elemental content will increase under nutrient replete growth, and (5) the effect of temperature on especially P and RNA content will differ between P-limited and nutrient replete growth. This is expected, because under P-limitation P and RNA more directly reflect the non-storage requirements, and P-content is not confounded by excess storage (luxury consumption), which may be significant under nutrient replete conditions (*Klausmeier et al., 2008*).

## MATERIAL AND METHODS

We carried out experiments with the three species *Emiliania huxleyi* (Lohmann) Hay & Mohler, 1967; *Chrysochromulina rotalis* Eikrem & Throndsen, 1999; and *Prymnesium polylepis* (Manton & Parke) Edvardsen, Eikrem & Probert 2011—all haptophytes in the class Coccolithophyceae (=Prymnesiophyceae). The specific strains were initially isolated from the outer Oslo fjord (Norway), and have been kept in culture at the University of Oslo for several years. We cultivated the algae in IMR 1/2 medium, which is the original IMR medium with nutrient levels reduced to 50% (*Eppley, Holmes & Strickland, 1967*) and 10 nM selenite added (*Edvardsen & Paasche, 1992*). We used filtered seawater (1.2 μm Whatman GF/C) and adjusted the medium to a final salinity of 30 psu. Finally, we post-filtered the medium through a 0.22 μm sterile filter, pasteurized it at 80 °C for 15 min, and stored it dark at 14 °C.

The experiment was designed as a long term, factorial setup with two P-treatments (P-limited and nutrient replete; explained below) and two temperatures (13 °C and 19 °C) in triplicates for three species. This yielded 36 experimental units. Prior to the experiment, the algae were cultivated in IMR 1/2 medium for approximately two weeks at 16 °C before being acclimated to their respective temperatures for seven days.

The P-limited cultures (hereafter referred to as "−P") were grown in IMR 1/2 medium reduced to 1 μM phosphate, resulting in a dissolved N:P ratio of 124:1 (mol:mol). We grew the −P cultures as batch cultures in semi-continuous mode. Every second day, a constant fraction of the culture volume was removed and replaced by fresh medium. The fraction removed was 50% for *E. huxleyi* and *C. rotalis*, and 40% for *P. polylepis*. Hence, the dilution factors (DFs) were $1/(1-0.5) = 2$ and $1/(1-0.4) = 1.667$, respectively.

With the semi-continuous mode, we ensured that cell densities eventually reached a quasi steady state which was limited by P due to the high N:P supply ratio (124:1). Here the state of the algal cells is the same in each dilution cycle, but will change somewhat over the inter-dilution growth period due to the periodic dilution with fresh medium. The high P treatment (hereafter referred to as "+P") received standard IMR 1/2 with 12.5 μM phosphate resulting in a dissolved N:P ratio of 10:1. To make sure that we had no nutrient limitation in the +P cultures, these were run as batch cultures in turbidostat mode, and diluted down to the same, low cell density every 2–3 days; 50,000 cells mL$^{-1}$ for *E. huxleyi* and *C. rotalis* and 100,000 cells mL$^{-1}$ for *P. polylepis*. This ensured that the cells were P-saturated and always growing exponentially.

All experiments were run in 40 mL nunclon filtercap flasks (Thermo Scientific, Waltham, MA, USA). The cultures received an irradiance of 170 μmol photons m$^{-2}$s$^{-1}$ from fluorescent white tubes, and the light: dark-cycle was set to 14 h:10 h. The cultures were maintained for >20 weeks, with regular samples for determination of cell numbers and cell size. Elemental content (C, N, P), RNA and alkaline phosphatase activity (APA) was measured as indices of nutrient status of the cultures.

Cell size (equivalent spherical diameter; ESD) was measured by a CASY electronic cell counter (Schärfe System GmbH, Rautlingen, Germany). Mean and median ESD were analyzed twice a week after the cultures had stabilized. During the first weeks, cell densities were estimated from CASY cell counts, but when the cultures had stabilized, we started monitoring cell densities by measuring absorbance at 660 nm (A$_{660}$) using a plate reader equipped with a spectrophotometer (BioTek Synergy MX; Winooski, VT, USA). A$_{660}$ was converted to cell numbers based on species- and treatment-specific calibration curves constructed from parallel measurements of A$_{660}$ and cells numbers in the CASY ($R^2 > 0.9$ for all except one calibration curve, where $R^2$ was 0.75).

For the +P cultures, specific growth rate was calculated as $\log(N_{t+\Delta t}/N_t)/\Delta t$, where $N_t$ was the cell density after dilution, $N_{t+\Delta t}$ the cell density at the end of the growth interval (before the next dilution), and $\Delta t$ the time interval between measurements (in days). For the −P cultures, growth rate was calculated as $\log(N_{t+\Delta t}/(N_t/DF))/\Delta t$, where $N_t$ was the cell density before dilution, DF the dilution factor (see above), and $N_{t+\Delta t}$ the cell density at the end of the growth interval.

Since the cultures (and growth rates) remained stable over time, we here report mean growth rates for each species and treatment. Cultures were kept at the same light:dark cycle (14:10), and samples were taken at the same time in the morning to ensure that cells were harvested from the same cell cycle phase.

Analysis of particulate C, N, and P was performed for all cultures at steady state. For C and N analysis, algae were collected on a GF/C filter and analyzed on a Thermo Finnegan EA

1112 series flash analyzer (Thermo Fisher Scientific, Waltham, MA, USA). Some fraction of particulate C in *E. huxleyi* could be allocated to the calcified structures. Hence in order estimate the amount of particulate organic carbon (POC) and particulate inorganic carbon (PIC), additional samples of *E. huxleyi* were treated with 2 M hydrochloric acid (HCl) to remove PIC (*Langer et al., 2009*). We did not find significant differences in cellular C with or without this treatment, and we thus have based the analysis on non-acidified samples as for the other two species. Particulate P was estimated by filtering cells onto a GF/C filter before soaking the filter in 10 mL of a 1% solution of potassium peroxydisulfate for 30 min at 120 °C, followed by colorimetric analysis in an autoanalyzer (Bran Luebbe, Norderstedt Germany).

RNA was measured three times during steady state using a modified version of the RiboGreen fluorescence protocol (Turner BioSystems Sunnyvale, CA, USA). From each culture, 1–4 mL was filtered onto nitrocellulose membranes (0.65 µm DAWP, Millipore), which were snap-frozen (in liquid N) in nuclease-free tubes. Prior to analysis, the tubes were added 120 µL of the extraction buffer (1% sarcosyl; Sigma Aldrich, St. Louis, MO, USA), and while still frozen, homogenized by ice-cold sonication (Branson Sonifier; Branson Ultrasonics, Branson, CT, USA) for two minutes. The samples were again put on ice, and diluted 1:5 with TE buffer (10 mM Tris-HCL, 1 mM EDTA, pH 7.5). Duplicates with 75 µL from each sample were produced in individual slots of a 96-well plate (655076; Greiner Bio-One, Monroe, NC, USA). The duplicates were inserted pairwise into columns after the first column, which was reserved for an RNA standard. Into each of the first duplicate columns a total of 20 µL RNase-free water (BRL1071; Gibco, Grand Island, NY, USA) was added. The other duplicates were then added 20 µL of 0.1% RNase A (A7973; Promega, Madison, WI, USA). Immediately after the RNase mixture was added, the well plate was heat incubated at 37 °C on a shaking table to ensure homogenous dispatch of the RNase, and digestion of RNA. After the incubation, 75 µL of 100 times diluted RiboGreen dye (R-11490; Molecular Probes, Eugene, OR, USA) was added to each well with an automatic 8-channel pipette. Finally, the sample plate was inserted into a plate reader (described earlier) and fluorescence of the samples measured using an excitation wavelength of 480 nm (20 nm bandwith) and an emission wavelength of 525 nm (20 nm bandwith). Since RiboGreen yields fluorescence from both RNA and DNA, we subtracted the fluorescence in the RNase-treated samples from the non-RNase treated samples to obtain the signal from RNA alone. Concentrations of RNA were obtained by calibrating the fluorescence of the unknown samples against the fluorescence of the known RNA standards. We converted the weight of total RNA to moles of ribonucleotide monophosphates (which each contains one P-atom) using the average molecular weight of the four different ribonucleotide monophosphates (339.5 g mol$^{-1}$).

Alkaline phosphatase activity (APA) may serve as an independent biomarker for P-limitation, and was analyzed along with RNA for all treatments and species. APA was analyzed by the CDP-star chemoluminescence method according to the protocol of *Wojewodzic et al. (2011)*. Samples were collected as for RNA and stored at −80 °C prior to analysis. For analysis, samples were put on ice and 0.3 mL of Triton X-100 (T8787; Sigma Aldrich, St. Louis, MO, USA) was added. Samples were sonified as in the RNA procedure
described above. The standards were then prepared by using AP type VII-S from bovine intestinal mucosa (P5521; Sigma Aldrich, St. Louis, MO, USA). The standard curve ranged from 2-100 pU of the AP, diluted with 1% Triton X-100. One unit (pU) is in this context the amount of enzyme that is required to hydrolyze 1 µM 4nitrophenyl phosphate per minute at pH = 9.8 and a temperature of 37 °C. After the preparation of the standards, 20 µL of both the standards and the samples were transferred to a Pyrophosphate-free 96-well plate (236105; Nunc, Rochester, NY, USA) and preserved on ice. Afterwards, 20 µL of 0.4 mM CDP-star was dispensed with an automatic 8-channel pipette to all the wells, and luminescence of the samples measured in a plate reader.

### Biomass normalization of C, N, P, RNA, and APA

Because we observed significant differences in cell size between the experimental treatments, we chose to normalize the molar amounts of C, N, P and RNA, and APA, by the total cell volume in each sample. The total cell volume (in liters) in a sample was calculated as $\left[\frac{4}{3}\pi\left(\frac{\varphi}{2}\right)^3\eta\right]10^{-15}$, where $\eta$ is the total number of cells in the sample and $\varphi$ the median ESD (in µm) of the cells in the samples for C, N, P, RNA and APA. The factor $10^{-15}$ converts from µm$^3$ to L. The resulting bulk cell volume-specific concentrations have units of moles (L cell volume)$^{-1}$. We will also present results of cell quota, i.e., the number of moles of C, N, P, and RNA per cell.

### Data analysis

All data analysis was done in R (*R Core Team, 2016*). We used principal component analysis (PCA) on log-transformed and standardized cell volume-specific concentrations of C, N, P, and RNA from all species to get an overview of the general trends in elemental content and RNA, and how the variables were related to P-regime, temperature, and species. To estimate the fraction of variance in the dataset that was related to these three factors, we used permutational multivariate ANOVA (the function *adonis* in the *vegan* package). Moreover, we tested whether any of the underlying gradients in elemental content and RNA were related to growth rate and the degree of P-limitation by including growth rate and APA as passive variables in the ordination (using the *envfit* function in the *vegan* package; *Oksanen et al., 2017*). To test for additive and interactive effects of P-regime and temperature on the different response variables within species, we used linear models on log-transformed data. Temperature and P-regime were both represented as factor variables with two levels each.

## RESULTS

### The response of growth rate and APA to P-regime and temperature

Within the three species, 84–94% of the variation in specific growth rate was explained by P-regime (calculated from model sums of squares), with the +P cultures growing significantly faster than the −P cultures ($p < 0.05$, Table 1). This was expected, since we used two different culturing modes (semi-continuous vs. turbidostat mode) to impose P-limited and nutrient replete growth. Temperature did not influence the measured growth rate in any of the −P cultures ($p > 0.05$), which was expected since we used the

**Table 1 Specific growth rates ($\mu$; per day) in the different experimental combinations of P-regime and temperature.** The standard deviation of the three replicates is given in parenthesis. Regardless of species, cultures grown at +P always had significantly higher $\mu$ than those grown at −P ($p < 0.001$). A significant ($p < 0.05$) temperature-effect within a given P-regime is marked with an asterisk.

|  | *E. huxleyi* | | *P. polylepis* | | *C. rotalis* | |
|---|---|---|---|---|---|---|
|  | 13 °C | 19 °C | 13 °C | 19 °C | 13 °C | 19 °C |
| P+ | 0.855 (0.032) | 1.045 (0.016)* | 0.59 (0.061) | 0.44 (0.018) | 0.605 (0.102) | 0.672 (0.015)* |
| P− | 0.36 (0.021) | 0.337 (0.011) | 0.186 (0.015) | 0.229 (0.042) | 0.407 (0.032) | 0.306 (0.009) |

same dilution rate at high and low temperature. High temperature increased growth rate in the +P cultures of *E. huxleyi* and *C. rotalis* ($p < 0.05$). In contrast, the +P cultures of *P. polylepis* grew slower at 19 °C compared to 13 °C ($p < 0.05$). APA was strongly elevated in the −P cultures for all species and temperatures, except for *C. rotalis* at 13 °C, where APA was similar for both P-regimes (Fig. S1 and Table S1; 58% of the variation in APA was explained by P-regime alone). Temperature had no significant additive effect on APA ($p > 0.05$), but there was a significant interaction between temperature and species ($p = 0.035$) manifested by elevated APA at 19 °C in P-limited *E. huxleyi*.

## Overall trends in C, N, P, and RNA

A PCA plot of the cell volume-specific concentrations of C, N, P, and RNA from all species revealed an underlying gradient from high to low concentrations along PC axis 1, which explained 78.7% of the variation in the dataset (Fig. 1). Cultures grown in +P clearly distributed themselves in the left part of the PCA plot along with high cell-specific concentrations of C, N, P, and RNA. The multivariate ANOVA supported this observation: P-regime explained 32.7% of the variation in the dataset, compared to 9.5% by temperature, and 2.8% by the interaction between P-regime and temperature ($p < 0.05$). Species differences accounted for 28%. Both specific growth rate and APA correlated strongly with the ordination ($R^2$ for growth rate $= 0.55$, $R^2$ for APA $= 0.41$; both $p < 0.01$). The passive variables are plotted as dashed arrows in Fig. 1, and each dashed arrow points in the direction to which the linear change in the variable is the fastest.

The corresponding analyses for cell quota of C, N, P, and RNA (Fig. S2 and Table S2) revealed the same general trends, but with a much stronger influence of species due to the large difference in cell size between the three haptophytes (average peak ESD for *E. huxleyii* $= 4.67$ μm [sd: 0.22 μm]; *C. rotalis*: 4.0 μm [sd: 0.1 μm] *P. polylepis*: 8.49 μm [sd: 0.55 μm]).

## Specific effects of P-regime

For elemental and RNA content (Figs. 2A–2C, Table 2), we found that cell volume-specific concentrations of P and RNA were lower under P-limitation for all species at both temperatures. The relative differences in P concentration between +P and −P was a factor 5.2 at 13 °C and 5 at 19 °C for *E. huxleyi*; 1.7 at 13 °C and 6.3 at 19° for *C. rotalis*; and 1.3 at 13 °C and 1.8 at 19 °C for *P. polylepis*. For C and N, the effect of P-regime was less consistent, but generally, the concentrations increased or were unaffected by P-regime (Fig. 2, Table 2). Due to the strong correlation between cell-volume specific concentrations

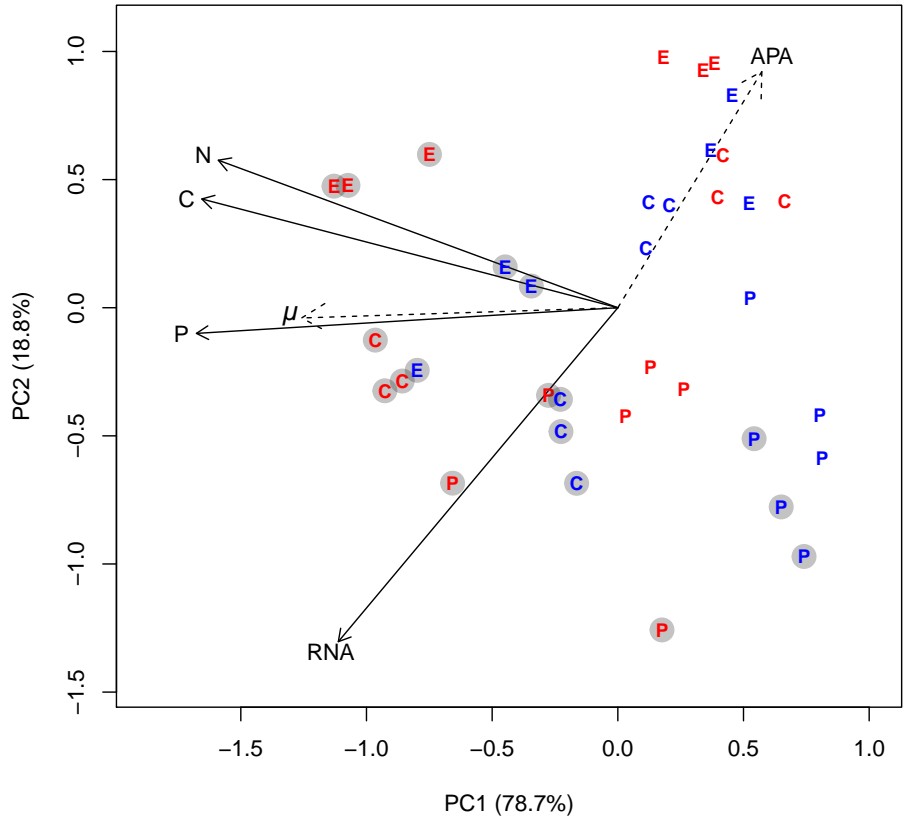

**Figure 1  PCA plot based on log-transformed and standardized cell volume-specific concentrations of C, N, P, and RNA in the 36 experimental units.** Due to the factorial design with three species, two temperatures, and two P-regimes, we obtained 12 unique experimental combinations which are represented in the diagram as follows: The letters E, P, and C represents the species *Emiliania huxleyi*, *Prymnesium polylepis*, and *Chrysochromulina rotalis*; the red and blue color represents 19 °C and 13 °C, respectively; and the grey circles represent the +P cultures. PC axis 1 accounted for 78% of the variation in the dataset, while PC axis 2 accounted for 18.8%. Specific growth rate ($\mu$; day$^{-1}$) and alkaline phosphatase activity (APA) were included as passive variables in the PCA plot (they did not influence the ordination, but were fitted to the ordination afterwards; see 'Methods') and plotted as dotted arrows. Each arrow points in the direction to which the linear change in the variable is the fastest.

of C and N ($r = 0.96$; all data pooled), C:N was not significantly affected by P-regime nor temperature (data not shown). For comparison, the correlations between C and P, and N and P were 0.92 and 0.85, respectively.

Cell *quotas* of C, N, P, and RNA were generally also higher in cultures growing at +P. However, for cell quotas the effect of P-regime was weaker and in more cases non-significant compared to the cell-volume specific concentrations C, N, P and RNA (Fig. S3 & Table S3). This can likely be explained by the overall trend of reduced cell size at +P (Fig. S4, Table S4).

## Specific effects of temperature

We found significant temperature effects on cell volume-specific concentrations of C, N, P, and RNA, but the effect differed between species, P-regime, and element (Fig. 2; *p*-values and coefficient estimates are found in Table 2). For all cultures growing at +P, cell volume-specific C, N, and P increased significantly with temperature. RNA increased

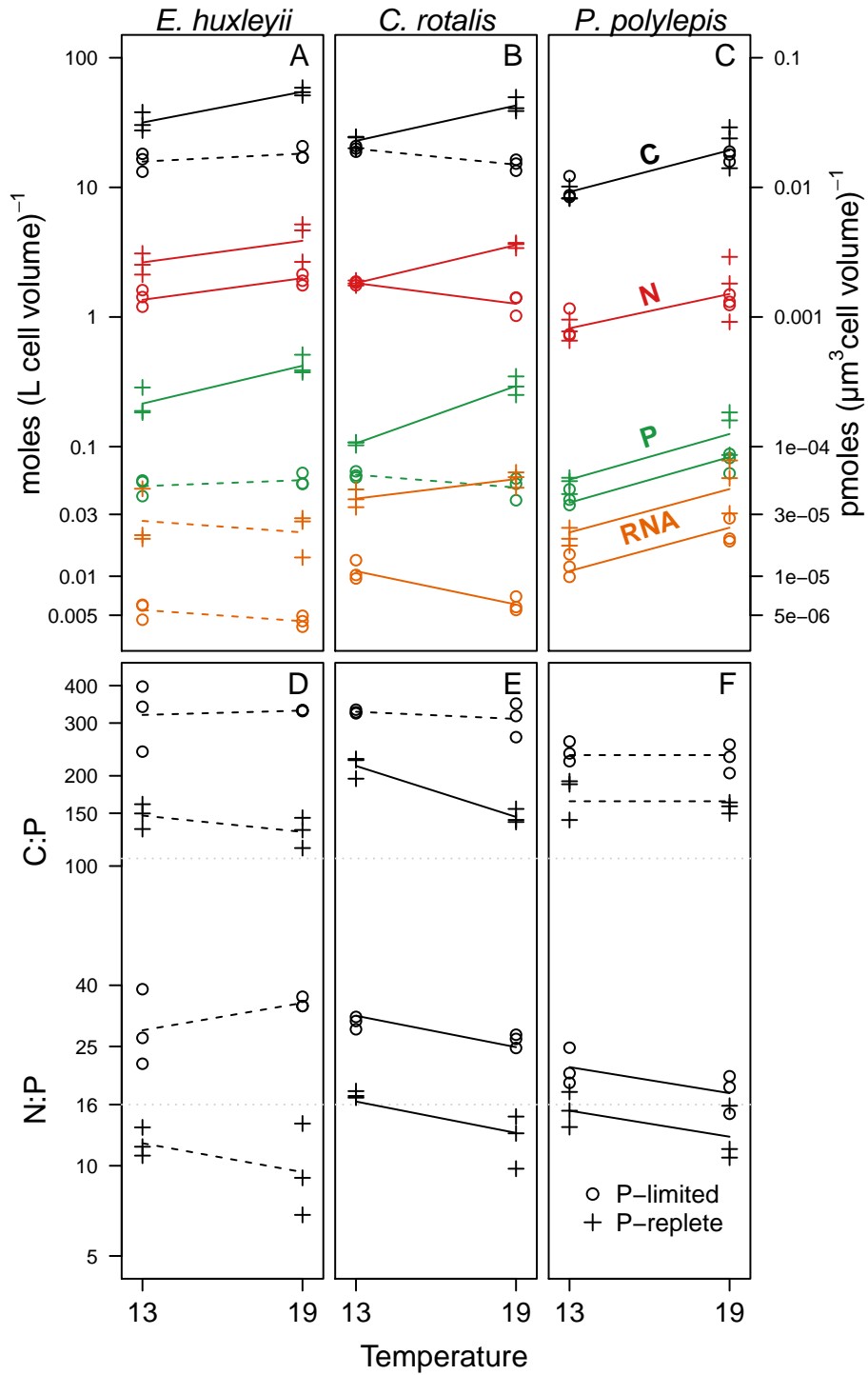

**Figure 2 Responses of C, N, P, RNA, C:P and N:P to temperature and P-regime within species. C, N, P, and RNA are expressed as moles per liter of cell volume (and pmoles per μm³ cell volume; right y-axis). Ratios are atomic. The data are plotted on log-scale. Dots represent P-limited cultures (−P), pluses P-replete cultures (+P).** (continued on next page...)
**Figure 2 (…continued)**
Solid lines are drawn if the difference between temperatures was significantly different from zero within a given P-regime (see Table 2 for coefficient estimates and *p*-values). Dotted lines indicate non-significant trends. If two lines are drawn for an element or a ratio, there was a significant difference between P-regimes for at least one of the two temperatures. If only one line is drawn (as for N in *P. polylepis*), no P-effect was present. Grey dotted lines show the Redfield C:P and N:P ratios.

**Table 2** Estimates and *p*-values from linear models on the form $\log_{10}(y) = \beta_0 + \beta_1 \times$ P-regime $+ \beta_2 \times$ temperature $+ \beta_3 \times$ P-regime: temperature, where *y* is either C, N, P, RNA, C:P, or N:P. P-regime and temperature are both represented as factor variables with −P and 13 °C as reference levels, respectively. Hence, $\beta_1$ is the estimated difference (on $\log_{10}$ scale) in y between +P and −P cultures growing at 13 °C, and $\beta_2$ the difference between 19 °C and 13 °C for cultures grown at −P. The interaction term $\beta_3$ represents the difference in temperature effect between +P and −P cultures. The last column shows the $R^2$ of the most adequate model, with the fraction explained by each significant term (P-regime, temperature, and the interaction between them) in parenthesis. $n = 12$ in all models.

| | $\beta_0$ (Intercept) | $\beta_1$ (P-regime) | $\beta_2$ (Temperature) | $\beta_3$ (P-regime: Temp) | $R^2$ (percentage explained by each term) |
|---|---|---|---|---|---|
| *E.huxleyi* | | | | | |
| C | 1.19[***] | 0.3[**] | 0.06 (NS) | 0.176[*] | 0.95 (79, 11, 4) |
| N | 0.13[*] | 0.289[**] | 0.167[*] | NS | 0.81 (61, 20, −) |
| P | −1.31[***] | 0.637[***] | 0.047 (NS) | 0.246[*] | 0.97 (90, 4.5, 2.4) |
| RNA | −2.26[***] | 0.684[***] | NS | NS | 0.88 (88, −, −) |
| C:P | 2.51[***] | −0.37[***] | NS | NS | 0.91 (91, −, −) |
| N:P | 1.5[***] | −0.47[***] | NS | NS | 0.86 (86, −, −) |
| *C. rotalis* | | | | | |
| C | 1.3[***] | 0.061 (NS) | −0.123[*] | 0.395[***] | 0.95 (57, 5, 33) |
| N | 0.261[***] | −0.00006 (NS) | −0.59[*] | 0.45[***] | 0.96 (46, 4, 46) |
| P | −1.21[***] | 0.24[**] | −0.099 (NS) | 0.54[***] | 0.98 (70, 9, 20) |
| RNA | −1.96[***] | 0.56[***] | −0.26[***] | 0.41[**] | 0.98 (91, 0.4, 7) |
| C:P | 2.51[***] | −0.18[**] | −0.024 (NS) | −0.146[*] | 0.96 (79, 11, 6) |
| N:P | 1.5[***] | −0.29[***] | −0.10[*] | NS | 0.92 (81, 11, −) |
| *P. polylepis* | | | | | |
| C | 0.98[***] | NS | 0.26[*] | NS | 0.77 (−, 77, −) |
| N | −0.87[***] | NS | 0.27[*] | NS | 0.53 (−, 53 , −) |
| P | −1.43[***] | 0.18[*] | 0.35[**] | NS | 0.82 (17, 65, −) |
| RNA | −1.96[***] | 0.298[*] | 0.334[*] | NS | 0.80 (36, 45, −) |
| C:P | 2.37[***] | −0.154[**] | NS | NS | 0.77 (77, −, −) |
| N:P | 1.33[***] | −0.145[*] | −0.087[*] | NS | 0.68 (50, 18, −) |

**Notes.**
Coding of *p*-values.
[***]<0.0001.
[**]<0.001.
[*]<0.05.

with temperature at +P two out of three species. Under P-limitation, we did not detect any consistent trend in elemental content or RNA with temperature: C and N increased with temperature in two of three species, while P and RNA decreased or did not change with temperature in two of three species.

Cell *quota* of C, N, P, and RNA typically responded less to temperature than the cell volume-specific concentrations, although the direction of the response in most cases was

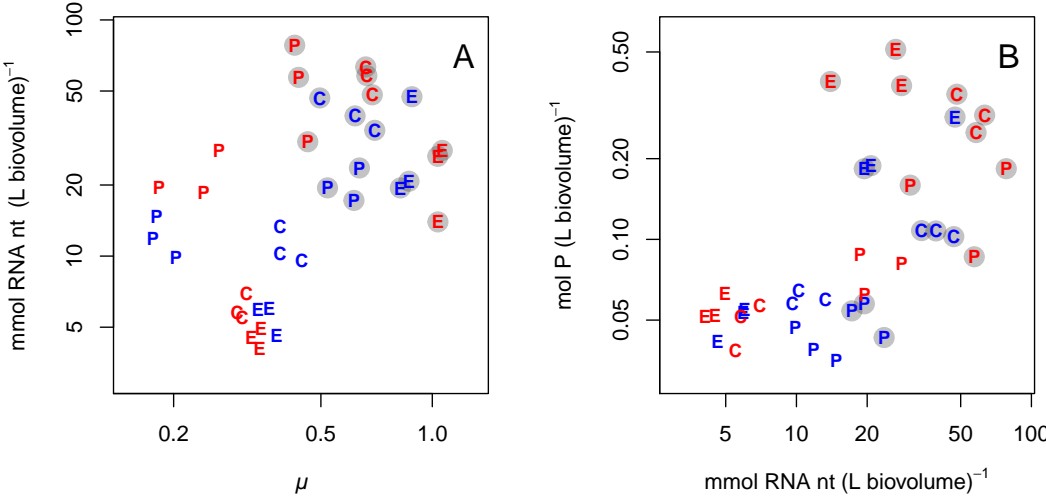

**Figure 3** (A) Cell volume-specific concentration of RNA (mmol RNA nucleotides (nt) per L biovolume) as a function of specific growth rate ($\mu$). (B) Cell volume-specific concentration of P as a function of RNA. Note log-transformed axes; dot symbols are as in Fig. 1.

similar (Fig. S3 & Table S3). The weaker signal of temperature on cell quota is likely linked to the general decrease in cell size with temperature (Fig. S4, Table S4).

The effect of temperature on C:P and N:P ratios was either negative or non-significant (Figs. 2D–2F, Table 2): For C:P, the only significant decrease with temperature was in the +P culture of *C. rotalis*. N:P, on the other hand, decreased significantly with temperature under both P-regimes for *C. rotalis* and *P. polylepis*. In *E. huxleyii*, N:P tended to increase with temperature under P-limitation and decrease under +P, but none of the trends were significant. For all species, C:P ratios were well above Redfield proportions (C:P = 106) under both P-regimes. N:P ratios were above Redfield (N:P = 16) for P-limited cultures, but below or close to Redfield when growing under nutrient replete conditions (Figs. 2D–2F).

## Growth rate, RNA, and P

Since RNA content was consistently higher for populations growing at +P, RNA content was also positively correlated with growth rate (Fig. 3A; all species pooled). Further, high P content coincided with high RNA content (Fig. 3B; all species pooled). In both these relationships there was a grouping of the P-limited treatments, which had low growth rate, RNA, and P content, and the +P treatments, which had high growth rate, RNA, and P content. Also, species grouped together to some extent. Due to the grouped structure of the data, we did not fit any statistical model and only present graphical relationships in Fig. 3.

The fraction of total cellular P accounted for by RNA-P (i.e., moles of ribonucleotide monophosphate per moles of total cellular P) in the three species ranged from 4 to 66% and was significantly related to temperature ($p < 0.001$), P-regime ($p = 0.025$), and species ($p < 0.001$; Fig. 4). In *E. huxleyii* and *C. rotalis*, the fraction was consistently higher at low

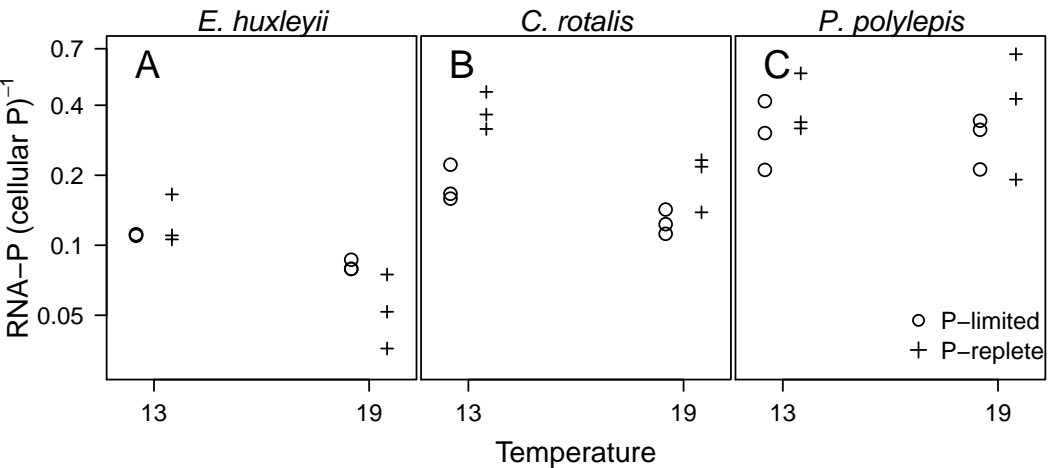

**Figure 4** **The fraction of P in RNA (i.e., moles of P bound in RNA: moles of total cellular P) plotted as a function of temperature and P-regime for each species.** Dots represent P-limited cultures (−P), pluses represent P-replete cultures (+P). Note the log-scale on the $y$-axis.

temperature, especially at +P, but this was not seen in *P. polylepis.* The effect of P-regime on the fraction of P in RNA was less consistent and varied with species and temperature.

## DISCUSSION

This study revealed significant and consistent effects of P-regime and temperature on elemental content and total RNA in three haptophytes. The main gradient in cell volume-specific concentrations of C, N, P, and RNA was largely related to P-regime and the resulting differences in growth rates and degree of P-limitation, the latter being independently confirmed by the striking differences in APA. For all species and temperatures, we observed lower cell volume-specific concentrations of P (and RNA) in the P-limited cultures compared to the exponentially growing nutrient replete cultures, which is consistent with classical Droop kinetics (*Droop, 1974*). Also, the concentrations of the non-limiting elements C and N were lower in most of the P-limited cultures, most notably at high temperature. Cell volume-specific concentrations of C and N were reduced less than P when comparing +P and P-limitation, which led to consistently higher N:P and C:P ratios in the P-limited cultures. That we observed no significant effects of neither P-regime nor temperature on the C:N ratio fits well with the general pattern for C:N ratios in phytoplankton cultures, which is reported to remain relatively constant unless N is limiting (*Geider & La Roche, 2002*).

Temperature influenced elemental and RNA content, but the responses partly contradicted our initial hypothesis, namely that low temperature should increase elemental content and RNA due to a higher allocation to nutrient rich macromolecules to compensate for reduced specific reaction rates at low temperature (*Rhee & Gotham, 1981*; *Woods et al., 2003*; *Reich & Oleksyn, 2004*; *Toseland et al., 2013*; *Yvon-Durocher et al., 2015*). Instead, we found lower cell volume-specific concentrations of P, N, and C (and RNA in two of three species) at low temperature in the +P cultures. In the P-limited cultures, however, there
was a tendency for a negative or a non-significant effect of temperature on P and RNA (but also N and C in some cases). This pattern was quite consistent in *E. huxleyi* and *C. rotalis*, but not in *P. polylepis*. A corresponding, positive effect of temperature on N and P content under nutrient replete conditions, but a negative effect under P-limitation (RNA decreased with temperature in both P-regimes), was also observed a similarly designed experiment with the freshwater chlorophyte *Chlamydomonas reinhardtii* (*Hessen et al., 2017*).

The tendency for opposite effects of temperature on elemental content and RNA in nutrient replete and P-limited cultures is quite striking, and could be related to differences in uptake and storage strategies of P in the two growth regimes. Under P-limitation, cells cannot store extra P, and must allocate the available P to essential, non-storage macromolecules that are required to sustain growth, e.g rRNA (*Sterner & Elser, 2002*). Hence, any changes in cellular P with temperature under P-limitation should reflect a change in P allocated to these essential, non-storage fractions of biomass. It is the allocation to rRNA that may be expected to change according to the translation efficiency hypothesis, which specifically states that cells should invest more in rRNA (and thus require more P) at low temperature. In fact, our observed elevated RNA at low temperature in two of three species under P-limitation is in support of this, as are the results from a corresponding experiment with *C. reinhardtii* (*Hessen et al., 2017*). Under nutrient replete conditions, however, cells may take up and store P in excess e.g as polyphosphates (*Powell et al., 2009*), which may overshadow any responses in the non-storage fraction of P. If, for example, the uptake rate of an element increases with temperature (more than the specific growth rate), higher temperature will lead to higher total cellular nutrient content even though the fraction of nutrient in the non-storage part of biomass has an opposite response to temperature. In *E. huxleyi* and *C. rotalis* (but not in *P. polylepis*) the fraction of total cellular P bound to RNA was higher at low temperature in both P-regimes. Under P-limitation, the changing fraction was due to a slightly more negative temperature response in RNA than in P. At +P, however, total cellular P increased more than RNA when moving from 13 °C to 19 °C. This points to an increased fraction of storage P at high temperature, perhaps due to more efficient uptake kinetics, which again lead to lower RNA:P.

The responses of cellular P and RNA to temperature in the P-limited cultures may also be related to differences in relative growth rate (RGR). The P-limited cultures of a given species were grown using the same semi-continuous dilution rate, which led to the same quasi steady state growth rate ($\mu$) at both 13 °C and 19 °C. Since the maximum growth rate ($\mu_{max}$) varied with temperature, the cultures also experienced differences in the RGR ($\mu$: $\mu_{max}$). Given that the maximum growth rate increased with temperature, the P-limited cultures at 19 °C experienced a lower RGR and were presumably more strongly P-limited than those growing at 13 °C. This would further be expected to result in lower P and RNA content (following Droop kinetics), which we observed to some extent in *E. huxleyi* and *C. rotalis*. In *P. polylepis*, we observed that cellular P and RNA clearly increased with temperature also under P-limitation. Interestingly, the maximum growth rate of *P. polylepis* was lower at 19 °C than at 13 °C, meaning that its RGR was higher at 19 °C.

We hypothesized that high temperature would be associated with high N:P ratio, but our experimental results did not support this. N:P in fact decreased with temperature in

most cases, except for in P-limited *E. huxleyi*. Our results thus run contrary to a recent meta-analysis of 14 experimental datasets with N:P vs. temperature from phytoplankton cultures (*Yvon-Durocher et al., 2015*). Here, the average response of N:P to temperature was positive within species, yet with large differences in effect between species and experimental units (*Yvon-Durocher et al., 2015*). In a recent study of six strains of *Prochlorococcus* cyanobacteria, temperature had a weakly significant positive effect on N:P and C:P, but the ratios decreased again as temperature surpassed the optimum temperature for growth (*Martiny et al., 2016*). Adding the results from our study to these data, it seems evident that species-specific effects, and likely also effects of growth regime (exponential vs. nutrient limited) and type of nutrient limitation, are important in determining the response of N:P to temperature.

Due to differences in cell size between treatments, we focused on cell volume-specific concentrations when addressing the effects of temperature and P-regime on elemental content and RNA. Comparing the responses of cell volume-specific concentrations and cell quota, we generally found weaker effects of both P-regime and temperature on the cell quota. This can likely be explained by the fact that cells were smaller at high temperature and +P, the two types of treatment that generally also were associated with higher cell volume-specific concentrations. Hence, cells became smaller, but also more nutrient-dense at high temperature and +P. Recent studies suggest that much of the effect of temperature on cell size may in fact be caused by increased nutrient limitation both within and between species (*Peter & Sommer, 2013*; *Marañón, 2014*). Nutrient availability dominates over temperature as a predictor for the size structure of phytoplankton communities in the ocean, with small species dominating in areas of low nutrient availability (*Marañón et al., 2012*). Our results on cell size indicated that temperature was the most important variable, temperature and does not support nutrient limitation as an important driver for intraspecific changes in the cell size of these haptophytes. In fact, if there was an effect of P-regime, cells were smaller at +P.

Any changes in the stoichiometry and size structure of natural phytoplankton communities with temperature will, to a large extent, be driven by shifts in species composition (*Martiny et al., 2013*; *Marañón et al., 2012*). To what degree intraspecific responses to temperature and nutrient limitation will be important is still an open question, but the results from this study indicate that the responses to temperature likely will depend on whether nutrients are limiting or in surplus. Hence, increasing temperature will likely affect phytoplankton in high latitude and nutrient rich waters differently from that of communities at low latitude and nutrient limited regions.

### Funding
The authors received no funding for this work.

### Competing Interests
The authors declare there are no competing interests.

## Author Contributions

- Lars Fredrik Skau performed the experiments, analyzed the data, contributed reagents/materials/analysis tools, wrote the paper.
- Tom Andersen conceived and designed the experiments, analyzed the data, contributed reagents/materials/analysis tools.
- Jan-Erik Thrane analyzed the data, wrote the paper, prepared figures and/or tables, reviewed drafts of the paper.
- Dag Olav Hessen conceived and designed the experiments, contributed reagents/materials/analysis tools, wrote the paper, reviewed drafts of the paper.

## Data Deposition

The raw data is included as a Supplemental Dataset.

## Supplemental Information

Supplemental information for this article can be found online at http://dx.doi.org/10.7717/peerj.3743#supplemental-information.

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
