# Peer review of "Growth, stoichiometry and cell size; temperature and nutrient responses in haptophytes"

_PeerJ, doi:10.7717/peerj.3743_

## Round 0.1 · original submission · Major Revisions

Dear authors,

The reviewers have recommended publication, but one of them suggested some major revisions.Please respond to the reviewers comments and revise your manuscript paying particular attention to the comment regarding the method section.

·

Basic reporting

no comment

Experimental design

no comment

Validity of the findings

no comment

Additional comments

Very nice paper which fits perfectly into the current development of the field.
I have only very view comments:

Line 93: delete “at“.
Line 117: Insert “is expected“ between “this” and “because”.
Line 160/161: Growth rate µ should not only be the rate of change through time but also include the dilutions rate.
Line 331: insert verb (e.g. “expected”, “reported”,….) between “is” and “to”.

Reviewer 2 ·

Basic reporting

Article is formally correct. English needs some polishing (for instance, prepositions seem to go missing often). Data are given in the appendix.

Experimental design

The experimental design has some problematic issues, which are described below in my comments to the authors. Also, the estimates of cell abundance based on fluorescence and of growth rates need attention. The description of the experiments needs to be revised so that the readers are not misled about the experimental setup used.

Validity of the findings

The findings are valid, but require some additional discussion, particularly with regard with the underlying physiological mechanisms. In addition, some methodological assumptions need to be supported with additional data (e.g. calibration between fluorescence and abundance)

Additional comments

This is a valuable and timely contribution, because it demonstrates the importance of considering the interaction between temperature and nutritional status to understand the variability in resource allocation and biochemical composition of phytoplankton. The main conclusion is that nutrient availability affects the response of phytoplankton resource allocation and biochemical composition to changes in temperature. There are, however, several important issues regarding the experimental setup, and the measurements of growth rates, that need to be addressed before publication can be recommended.
Major comments
I have three major comments regading the methods used, and a couple of suggestions to improve the Introduction and the DIscussion.
1. Authors refer to their cultures as steady-state cultures (e.g. line 143), but they were not. This is particularly important in the case of the low-P cultures: P-starved cells will take up phosphate very fast following dilution with fresh medium, and increase their P quotas, which will then go down until next dilution takes place. In this setting, no steady state can ever be achieved, and growth remains unbalanced. This affects also pigment content, and hence fluorescence (see main comment 2 below). It also affects critically the estimates of growth rate in low-P cultures (comment 3 below).
Similarly, the high-P cultures were diluted every 2-3 days: they were not turbidostats, because in turbidostats biomass is removed constantly so that continuous growth is achieved. Surely cells divide over a period of 2-3 days, hence abundance cannot be constant, as authors state on line 146. In summary, authors should clarify that they were running batch cultures in semi-continuous mode, and avoid terms such as 'steady-state', 'turbidostat' or 'continuous cultures' when describing their experiments.
2. Estimates of abundance/biomass made with fluorescence pose problems. This is because both the cellular content of chla and the fluorescence yield of chla are very sensititive to nutritional status. The authors should give the statistics of the calibration between fluorescence and cell abundance and, if possible, show the parallel data of fluorescence and cell abundance from those first weeks of experiment. This is required to ensure the readers that the estimates of growth rate are robust.
3. Estimates of growth rates in P-limited cultures are difficult to interpret (and compare with those of the P-sufficient cultures), because they reflect both rate (Blackman) and yield (Liebig) limitation. Given that PO4 concentration was very low, shortly after dilution growth rates will start to slow down simply because there is no more P available (yield limitation). This is different from a situation in which a slow growth rate is maintained by a persistent low rate of nutrient supply (Blackman limitation). To help address this issue, it would be helpful if the evolution of cell abundance (or fluorescence) is shown over time for the various consecutive dilution cycles.
I would suggest the authors to discuss the growth rate hypothesis (Sterner & Elser 2002) in the Discussion section in connection with the different response in P content with temperature in P-sufficient and P-limited cultures. While the mechanism suggested by the authors (the overshadowing effect of storage P) is plausible, another possibility is that a warmer temperature, which results in faster growth rate in P-sufficient cultures, leads to a faster pace of protein synthesis and hence more ribosomes and a higher P content.
The other suggestion is to place the P-limited cultures in an oceanographic context, referring in the first paragraph of the Introduction to the fact that in large expanses of the ocean phytoplankton growth is nutrient-limited (Moore et al. 2013 Nature Geoscience), hence the interaction between temperature and nutrient status is of considerable ecological and biogeochemical importance.
Specific comments
line 64 Note this mechanism is not universal. In polar regions, warming (plus freshening) will increase stability and reduce light limitation, which together with less ice coverage is expected to enhance productivity.
82 In the study of Rhee and Gotham, it is unclear if the increase in cellular N and P at low temperatures was indeed an increase on a volume basis - it may have just been a result of larger cell size at low temperatures.
98-99 There is also geographical variability in the relative contribution of heterotrophic bacteria, detritus and phytoplankton to total suspended particulate matter. For instance, detritus, which is P-poor (as P is recycled very fast), contributes more to total POM in oligotrophic regions than in coastal regions. This can also help explain the higher N:P ratio in the former. Another mechanism that works to cause higher N:P ratios in oligotrophic regions is that nutrient-starved cells can replace P-rich molecules such as membrane phospholipids with non-P-containing lipids (Van Mooy et al. 2009 Nature), while there is smaller physiological plasticity to replace N-rich molecules.
106-107 Difficult but not impossible. Maranon et al. 2012 L&O have shown that certain areas in the ocean have high nutrient availability and high temperatures, while nutrient limitation in cold waters is also possible (HNLC regions in the southern ocean). Nutrients, or resources more generally, vastly dominate over temperature as drivers of phytoplankton size structure.
134 It would be helpful to indicate here how these temperatures fit within the thermal growth curve of these spp, e.g. might 19ºC be supra-optimal for any of the species, or are both temps within the linearly (or exponentially) increasing part of the growth curve? If species have different Topt and thermal ranges, it could help explain some of the differences observed in biochemical composition responses to temperature.
140-141 Frequency of dilution should be indicated. One can achieve a dilution rate equivalent to 0.5 d-1 diluting 1:1 every day, or diluting 1:3 (culture:fresh medium) every 2 days. Frequency of dilution affects the temporal variability of cellular biochemical composition.
172 So was this a non-calcifying strain of Ehux?
Minor comments
45 'Contrary to'
49 'phytoplankton' is a plural noun, so 'phytoplankton constitute...'
62 'ecology of'
68 'respond'
79-82 The study of Chrzanowski and Grover 2001 L&O is also relevant here
117. 'This is because...'
119 'confounded by'
Figs 1, 3 Grey circles are barely visible, should be replaced by bold-line circles.
Fig. 2 Data would be easier to compare with values in the literature if units were pmol um-3.

Reviewer 3 ·

Basic reporting

no comment

Experimental design

no comment

Validity of the findings

no comment

Additional comments

This article is excellently written and professionally structured with good use of literature and sufficient context provided. Overall, an excellent piece of work and well presented data and figures. I thoroughly enjoyed this paper and commend the authors on their professionalism and thoroughness.

I have only minor comments and some required editing:

Line 117: this sentence is a little unclear and could benefit from a minor re-structuring, I'm not entirely sure what the authors mean by 'confounded' to excess storage.

In the discussion the authors refer to non-storage and storage fractions of biomass. For example on line 350, the sentence describes allocation of P to non-storage fractions and continues to say 'these macromolecules', but it is unclear to which macromolecules the authors are referring. A little bit more detail would help to clarify this issue.

Also, I couldn't see any figure legends for the supplementary figures. This should be included.

Super minor comments:
Line 45: add 'to' after Contrary
Line 59: delete the 's' in constitutes and also in line 60, change 'is to 'are', as phytoplankton is plural not singular, Word seems to get this wrong all the time.
Line 62: add and 'of' between ecology -- phytoplankton
Line 93: delete 'at'
Line 117: add an 'is' after This.
Line 118: delete the s from reflect
Line 180: add 'to' before the tubes
Line 203: delete 'in'
Line 204: move 'added' to after parentheses
Line 331: add 'in' before phytoplankton cultures
Line 365: add and 's' to point
Line 399: It is unclear what the authors mean by the 'most important driver at the intraspecific level', please elaborate a bit more here.
Line 406: delete the 's' in indicates.

---

## Round 0.2 · accepted · Accept

Dear Authors,

Thank you for your revision which I have evaluated. The article is now accepted for publication.

Thank you for your contribution